# Severe Hyporesponsiveness to Erythropoiesis-Stimulating Agents in Patients on Chronic Hemodialysis—Reconsidering the Relationship with Thrombo-Inflammation and Oxidative Stress

**DOI:** 10.3390/diagnostics14212406

**Published:** 2024-10-29

**Authors:** Srdjan Nikolovski, Branislava Medic Brkic, Katarina Savic Vujovic, Ivana Cirkovic, Nina Jovanovic, Bhavana Reddy, Omer Iqbal, Chongyu Zhang, Jawed Fareed, Vinod Bansal

**Affiliations:** 1Department of Pathology and Laboratory Medicine, Loyola University Medical Center, Maywood, IL 60153, USA; oiqbal@luc.edu (O.I.);; 2Institute for Pharmacology, Clinical Pharmacology and Toxicology, University of Belgrade School of Medicine, 11000 Belgrade, Serbia; 3Institute for Microbiology and Immunology, University of Belgrade School of Medicine, 11000 Belgrade, Serbia; 4Eye Institute “Knezovic”, 10000 Zagreb, Croatia; 5DeBusk College of Osteopathic Medicine, Lincoln Memorial University, Harrogate, TN 37752, USA; 6Department of Molecular Pharmacology and Neuroscience, Loyola University Medical Center, Maywood, IL 60153, USA; 7Department of Nephrology, Loyola University Medical Center, Maywood, IL 60153, USA

**Keywords:** end-stage renal disease, hemodialysis, anemia, erythropoietin, erythropoiesis-stimulating agents, erythropoietin hyporesponsiveness index, body mass index, L-type fatty acid binding protein

## Abstract

Background/Objectives: Besides a multitude of consequences patients on chronic renal replacement therapy have, anemia is one of the most prominent factors making a significant number of patients dependent on erythropoiesis-stimulating agent (ESA) therapy. The aim of this study was to examine the relationship between the levels of a broad spectrum of thrombo-inflammatory and oxidative stress-related biomarkers and the presence and level of ESA hyporesponsiveness in patients undergoing regular chronic hemodialysis. Methods: This cross-sectional study included 96 patients treated with chronic hemodialysis. Levels of several thrombo-inflammatory and oxidative stress-related biomarkers, as well as demographic, clinical, and laboratory analyses, were collected and analyzed based on the calculated value of the ESA-hyporesponsiveness index (EHRI). Results: In the analyzed sample, 58 patients received ESAs. Of all the investigated parameters, only body mass index (BMI), level of plasminogen activator inhibitor-1, and level of L-type fatty acid binding protein (L-FABP) were observed as significant predictors of EHRI. A significant diagnostic potential for ESA resistance has been observed in BMI and L-FABP between ESA-resistant and ESA-non-resistant groups of patients (*p* = 0.004, area under the curve 0.763 and *p* = 0.014, area under the curve 0.712, respectively) with the cut-off values of 25.46 kg/m^2^ and 5355.24 ng/mL, respectively. Having a BMI of 25.46 kg/m^2^ or less and an L-FABP level higher than 5355.24 ng/mL were observed as significant predictors of ESA resistance (odds ratio 9.857 and 6.125, respectively). Conclusions: EHRI was positively predicted by low BMI and high levels of plasminogen activator inhibitor-1 and L-FABP. High levels of L-FABP and low BMI have been observed as strong predictors of ESA resistance.

## 1. Introduction

End-stage renal disease (ESRD) represents a major public health problem globally and is associated with considerable morbidity and mortality [1,2]. The estimated prevalence of ESRD is expected to rise over the next decades, driven by population aging and the increasing prevalence of diabetes mellitus and hypertension [3,4].

According to the 2022 Annual Data Report of the United States Renal Data System, the incidence of ESRD is constantly rising in the United States [5]. The fact that only one-fifth of patients undergo kidney transplant procedures contributes even more to the fact that this condition represents a significant burden to healthcare systems worldwide [5,6].

ESRD patients often experience chronic inflammation due to the uremic milieu, elevated levels of proinflammatory cytokines, and oxidative stress. Among those patients, the presence of an inflammatory state may be closely related to accelerated atherogenesis, protein-energy wasting, and anemia [7,8,9]. Also, there is a chronic activation of the coagulation cascade in ESRD, leading to elevated levels of thrombin-antithrombin complexes and decreased levels of natural anticoagulants like protein C and protein S [10].

Given the increasing prevalence of ESRD, early identification of patients with this disease can improve its management and prognosis. Several biomarkers have been investigated in relation to these important points of treating ESRD patients, such as C-reactive protein (CRP), tumor necrosis factor-alpha (TNF-α), interleukin (IL)-1, IL-6, D-dimer, fibrinogen, tissue factor, von Willebrand factor, plasminogen activator inhibitor-1 (PAI-1), thrombin activatable fibrinolysis inhibitor (TAFI), tissue plasminogen activator, microparticles, and anti-platelet factor 4 [11,12,13,14,15,16].

Parallel with kidney dysfunction, renal anemia manifests in ESRD patients primarily due to impaired erythropoietin (EPO) production by specialized peritubular cells [17,18,19,20]. However, in patients on hemodialysis (HD), anemia is related to other factors, such as hemorrhages, impaired erythropoiesis, or oxidative damage of red blood cells. The main cause of anemia in patients undergoing regular HD is EPO deficiency, and the main clinical consequences are described as the development of cardiovascular diseases, decreased quality of life, and increased cardiovascular mortality risk [21]. Despite the administration of recombinant human erythropoietin (rHuEPO) as an erythropoiesis-stimulating agent (ESA) at appropriate doses, about 5–10% of patients treated with regular HD develop ESA resistance/hyporesponsiveness [22].

The aim of this study was to determine the levels of a broad spectrum of thrombo-inflammatory, oxidative stress-related biomarkers, as well as their association and causal relationship with the presence and level of ESA resistance in ESRD patients treated by regular chronic HD.

## 2. Materials and Methods

### 2.1. Study Population and Laboratory Data

In this cross-sectional cohort study, whole blood samples were drawn from 96 adult ESRD patients (estimated glomerular filtration rate less than 15 mL/min) treated with a regular HD program lasting at least six months. Before the initiation of the study, written informed consent from each participant was obtained, according to Loyola University Chicago’s Institutional Review Board approval (Institutional Review Board number 107346071204; Date of Approval: 12 July 2004). The study was performed in conformity with the guidelines of the Health Insurance Portability and Accountability Act of 1996, General Data Protection Regulation, and in agreement with the ethical standards laid down in the 1964 Declaration of Helsinki.

### 2.2. Samples and Data Collection

Plasma samples were collected in 3.8% (0.109 mol/L) sodium citrate tubes, processed for platelet-poor plasma, and stored at −70 °C prior to analysis. For comparison, control plasma samples from 40 healthy, non-smoking adults matched by age and gender, were processed and analyzed. The levels of D-Dimer, CRP, PAI-1, functional PAI-1, TAFI, tissue plasminogen activator, von Willebrand factor, fibroblast growth factor 23, IL-6, TNF-α, vascular endothelial growth factor, anti-platelet factor 4 immunoglobulin G, endogenous glycosaminoglycans, microparticles, ADAMTS-13, angiopoietin-2, pro-B-type natriuretic peptide, L-type fatty acid binding protein (L-FABP), lipopolysaccharide, non-esterified fatty acids, alpha-fetoprotein, myeloperoxidase, and nitrotyrosine were quantified by using commercially available enzyme-linked immunosorbent assay kits and chromogenic methods for each of the ESRD and control plasma samples. Clinical information, including patient demographics, comorbid conditions, laboratory results, and pharmacotherapy, was collected through a review of patient electronic medical records. The adequacy of hemodialysis was assessed by the Kt/V ratio and urea reduction ratio.

The data regarding ESA and iron application in patients were also obtained from medical e-charts and were recorded along with hemogram values. Information regarding frequency, dose, and type of iron supplementation was collected for all the patients receiving oral or parenteral iron supplementation.

The levels of thrombo-inflammatory and oxidative stress-related biomarkers were compared to the value of EHRI determined as a weekly dose of EPO per kilogram of body weight divided by hemoglobin level (g/dL) [23,24]. The patients were also divided into two groups, one of them being a group of patients having EHRI above the cut-off value represented by the 75th percentile (i.e., within the fourth quartile—ESA-resistant group) and the other being a group of patients with EHRI below the 75th percentile (i.e., within the first three quartiles of the distributed data—ESA-non-resistant group). We differentiated the terms “hyporesponsiveness” and “resistance”, defining ESA resistance as severe ESA hyporesponsiveness.

### 2.3. Statistical Analysis

Distribution of continuous variables was determined by Kolmogorov-Smirnov test with Lilliefors correction followed by graphic evaluation. Continuous data were described as mean ± standard deviation (SD) or median and interquartile range (IQR), as appropriate. Categorical data were presented as proportions (frequencies and percentages). Between-group association was assessed with independent sample t-test and Mann-Whitney U test, as well as with cross-tabulation tests—Chi-square test and Fisher’s exact test. The degree and level of correlation between the variables were tested with Spearman’s rho test. Linear and binary logistic regression were used to analyze the predictive potential of all the analyzed variables to the level of ESA resistance. Receiver operating characteristic analysis was used to determine the diagnostic power of ESA resistance predictors, while multilayer perceptron artificial neural network modeling was used to determine the importance level of the ESA resistance predictive model. The level of statistical significance was set to 0.05. The analysis was performed by using SPSS for Windows v27.0 (IBM Corp, Armonk, NY, USA) and GraphPad Prism v10 (GraphPad, La Jolla, CA, USA).

## 3. Results

This study included 96 patients undergoing regular HD for at least six months prior to blood sampling. In this group, 73 patients (76.0%) fulfilled the criteria for anemia secondary to chronic kidney disease (CKD), of which 58 were chronically receiving ESA therapy, for which further analysis was performed (Figure 1).

Biomarker levels in the ESRD group and control group are presented in Table 1. Significant differences between ESRD and the control group were observed in the levels of the majority of the analyzed biomarkers.

Within the group of patients receiving ESA therapy, 28 (48.3%) were males. Nine patients (15.5% were of Caucasian race, 33 (56.9%) were African-American, and 16 (27.6%) were of Hispanic origin. The median age was 67.0 (IQR 50.8–74.0) years. Twelve patients (20.7%) previously had a kidney transplant procedure (one of them was a recipient of two transplant procedures) followed by graft failure and/or rejection. Thirty patients (51.7%) were active smokers.

Median EHRI was 13.1 (IQR 5.3–21.4). All of the patients receiving ESA therapy received a short-acting type of rHuEPO and had hypertension, which was also the cause of CKD. The median Charlson comorbidity index was 6.0 (IQR 4.8–8.0). Chronic pulmonary disease was present in 25 patients (43.1%), ischemic heart disease in 21 (36.2%), chronic heart failure in 24 (41.4%), diabetes mellitus in 40 (69.0%), malignancy in 10 (17.2%), hypercholesterolemia in 6 (10.3%), and hyperlipidemia in 17 (29.3%).

Twelve patients (20.7%) were on chronic anti-inflammatory therapy (e.g., corticosteroids, immunosuppressants), 36 (62.1%) were chronically taking anti-phosphate agents, 33 (56.9%) were taking beta-blockers, 14 (24.1%) were taking angiotensin-converting enzyme inhibitors and/or angiotensin receptor blockers, 36 (62.1%) were on statins, and 14 (24.1%) were on anticoagulants.

Twenty patients receiving ESA therapy (34.5%) were taking iron supplementation, 29 (50.0%) were taking folic acid supplements, while 15 (25.9%), 28 (48.3%), and 26 (44.8%) were taking vitamin B12, vitamin C, and vitamin D supplements, respectively.

The median time patients receiving ESA therapy spent on a chronic HD program prior to the beginning of the study follow-up period was 50.0 (IQR 17.0–97.3) months. The mean body weight gained between HD sessions was 4.07 ± 2.62 kg, while the median Kt/V value was 1.53 (IQR 1.39–1.81). The mean values of body mass index (BMI) and body surface area before the HD session were 29.82 ± 7.84 kg/m^2^ and 1.95 ± 0.33 m^2^, respectively. The value of mean arterial pressure before the HD session was 91.41 ± 12.92 mmHg, while the median urea reduction ratio was 75.0 (IQR 70.1–77.3) %.

The median estimated glomerular filtration rate was 5.5 (IQR 4.8–6.3) mL/min/1.73 m^2^. Levels of serum iron and iron saturation were 57.0 (IQR 47.0–75.0) μg/dL and 27.0 (21.5–33.0) %, respectively. The mean values of serum transferrin and ferritin levels were 160.30 ± 26.59 mg/dL and 722.42 ± 351.71 ng/mL, respectively. The mean values of red blood cell count, hemoglobin levels, and hematocrit were 3.18 ± 0.41 × 1012/L, 9.98 ± 1.10 g/dL, and 31.56 ± 3.48%.

Distributions of clinical, demographic, and laboratory data, data on the distribution of comorbidities, as well as the data related to the applied pharmacotherapy in patients in the first three quartiles of EHRI (EHRI < 21.4, ESA-non-resistant group) and in the fourth quartile of EHRI (EHRI ≥ 21.4, ESA-resistant-group) are presented in Table 2, Table 3 and Table 4. Compared to the ESA-non-resistant group, parameters significantly lower in the ESA-resistant group were BMI, serum iron levels, and iron saturation, while neutrophil-to-lymphocyte ratio (NLR) and monocyte-to-lymphocyte ratio (MLR) were significantly higher in the same group (Table 2). No significant differences between the two groups were observed in the distribution of the analyzed comorbidities (Table 3). All of the patients taking angiotensin-converting enzyme inhibitors and/or angiotensin receptor blockers were in the ESA-non-resistant group (Table 4).

The distribution of the levels of investigated thrombo-inflammatory and oxidative-stress-related biomarkers in patients in the ESA-non-resistant group and in the ESA-resistant group is presented in Table 5. Of all the analyzed biomarkers, only ADAMTS-13 was significantly lower in the ESA-resistant group of patients.

All the investigated parameters were tested by linear regression analysis for their predictive potential for EHRI. Out of all the analyzed parameters, including clinical, demographic, and laboratory characteristics, presence of comorbidities, applied pharmacotherapy, and blood biomarkers, only BMI, PAI-1, and L-FABP showed significant predictive potential for EHRI, with the first one being negative and the other two being positive predictors for EHRI value (Figure 2).

The three biomarkers showing predictive potential for rhu–EPO resistance were further analyzed by multilayer perceptron artificial neural network modeling, with a training set consisting of 78.1% of patients and a testing set consisting of 21.9% of patients. BMI had the strongest importance of 0.498, followed by L-FABP with an importance level of 0.406 and PAI-1 having an importance level of 0.096. The normalized importance for BMI, L-FABP, and PAI-1 were 100%, 81.5%, and 19.2%, respectively. The analyzed model showed an area under the curve (AUC) of 0.878 (Figure 3 and Figure 4).

The same variables showing predictive potential in the linear regression analysis were assessed with receiver operating characteristic analysis. A significant diagnostic potential for ESA resistance was observed for BMI and L-FABP only (*p* = 0.004 and *p* = 0.014, respectively), while PAI-1 did not show a diagnostic potential (*p* = 0.983). The AUC for BMI as a diagnostic tool had a value of 0.763 (95% CI 0.582–0.943) with a sensitivity of 0.750 and specificity of 0.767 for the cut-off value of 25.46 kg/m^2^. Also, the AUC for L-FABP as a diagnostic tool had a value of 0.712 (95% CI 0.543–0.881) with a sensitivity of 0.636 and specificity of 0.778 for the cut-off value of 5355.24 ng/mL. These cut-off values for BMI and L-FABP were used to create two groups of patients having values of BMI and L-FABP lower and higher than the cut-off values. The proportion of ESA-non-resistant and ESA-resistant patients in both of the newly created groups of BMI and L-FABP was significantly different (Table 6).

Having BMI of 25.46 kg/m^2^ or less and with L-FABP level higher than 5355.24 ng/mL were observed as significant predictors of ESA resistance (odds ratios 9.857 and 6.125, respectively), compared to the cases when BMI was higher than 25.46 kg/m^2^ and when L-FABP levels were equal to or lower than 5355.24 ng/mL (Figure 5).

The artificial neural network model of all of the investigated biomarkers was analyzed separately in predicting ESA resistance, with a training set consisting of 71.4% and a testing set of 28.6% of patients, L-FABP had the highest importance level. When compared to its normalized importance of 100%, the following two biomarkers with the highest importance level were TAFI, with a normalized importance of 63.7%, and vascular endothelial growth factor with a normalized importance of 61.8% (Figure 6).

## 4. Discussion

The present study analyzed the broad spectrum of demographic, clinical, and laboratory data, comorbidities, and pharmacotherapy, as well as levels of several thrombo-inflammatory and oxidative stress-related biomarkers in patients with ESRD and renal anemia, treated with chronic HD and regular administration of rHuEPO. It evaluated the presence of a causal-consecutive relationship between these data and EHRI and compared them between the ESA-non-resistant and ESA-resistant groups of patients.

Studies have previously shown a multitude of inflammatory-related and oxidative stress-related factors associated with CKD, its pathophysiology, and the pathophysiology of its consequences [25]. Also, several parameters have been found in the previously published literature to have an association with ESA hyporesponsiveness in CKD patients [26].

Out of all investigated thrombosis-related biomarkers, we did not find differences between the two ESA resistance groups of patients. Levels of D-dimer, as one of the most prominent biomarkers for thrombosis conditions, were also similar between ESA-non-resistant and ESA-resistant groups. No significant findings exist in the literature regarding either D-dimer or EPO use, although one of the early studies showed that D-dimer increased significantly following EPO withdrawal [27].

In the search for potential predictors of hyporesponsiveness to EPO, Ingresciotta et al. [28] observed patient sex, age, baseline hemoglobin value, baseline ESA dosage, type of ESA used, category of hospital discharge diagnosis within one year prior to baseline date, comorbidities present, CKD stage or type of tumor, concomitant pharmacotherapy, and laboratory values as predictors of ESA hyporesponsiveness, highlighting serum CRP and high levels of baseline hemoglobin to be associated with poor response to ESA therapy. However, defining ESA hyporesponsiveness was unique due to the fact that it was assessed by calculating changes in hemoglobin levels over time. Additionally, that study not only included patients undergoing chronic HD but also cancer patients requiring chronic ESA use. In our study, we did not follow hemoglobin and other regular laboratory parameters throughout time due to a potential effect of subsequent ESA doses and due to the fact that all of the participants in this analysis were receiving ESAs before the blood sampling occurred.

A 2021 study conducted by Osman et al. [29] evaluated the impact of gender, laboratory values, BMI, and duration of HD on ESA resistance index in maintenance HD patients and found that female gender, low BMI, and high CRP levels as a marker of inflammation contributed to ESA hyporesponsiveness. Our study did not observe this kind of difference.

We observed only three differences in hemogram-derived cellular indices between the ESA resistance groups. Namely, only NLR, platelet-to-lymphocyte ratio, and MLR were significantly higher in the ESA-resistant group. This corresponds to previous findings [30,31]. However, some of the studies found the potential of platelet-to-lymphocyte ratio in predicting ESA therapy response [32], which was not observed in our study.

In this study, we observed lower BMI as a predictor of ESA resistance. Similar findings have already been shown previously [33]. Roldao et al. [34] found an association between ESA hyporesponsiveness and malnutrition but also a relationship with inflammation level, iron deficiency, and intact parathyroid hormone level, which we did not observe.

Previous studies have also shown that treatment with renin-angiotensin-aldosterone system blockers, inflammation, and secondary hyperparathyroidism could be considered predictors of ESA hyporesponsiveness [35]. In our study, patients in the ESA-resistant group used angiotensin-converting enzyme inhibitors/angiotensin II receptor blockers significantly less often.

We also did not observe differences in Kt/V values nor urea reduction ratio values as parameters of HD success rate between ESA-resistant and ESA-non-resistant groups, although some of the previous studies have found such a difference [33].

In the present study, as expected, CRP levels were significantly higher in the ESRD group, compared to controls. However, the presented level of inflammation was not higher in ESA-resistant patients. Both systemic immune-inflammation index and systemic inflammation response index were not significantly associated with ESA resistance in our study, which corresponds to the fact that CRP was also not different between the two groups. A number of studies have, however, previously shown the association between EPO dose/resistance, and increased CRP levels [36,37,38,39]. Some of them even found a cross-association with other anemia-related parameters. Locatelli et al. [40] reported lower CRP levels in ESA-hyporesponsive patients whose ferritin levels were higher, suggesting that ESA hyporesponsiveness is more prevalent in the presence of acute-phase response. We also did not observe similar findings in patients with normal or in those with high ferritin levels.

Two factors this study found to be the most significant for predicting ESA resistance were low BMI and high L-FABP levels.

Previous findings have described inflammation as an important contributor to the development of malnutrition and ESA resistance in hemodialyzed patients and have found a connection between inflammatory cytokines and ESA resistance level, which was not the case in our study. Feret et al. investigated a spectrum of cytokines (IL-6 and TNF-α, among others) and metabolic parameters as a determinant of malnutrition-inflammation syndrome (MIS) and their association with the level of ESA resistance [41]. In this study group, leptin as an adipokine was negatively correlated with both EHRI and MIS scores while positively with body weight, BMI, body surface area, and the proportion of adipose tissue in the body. Overall, patients with obesity had higher leptin levels and lower EHRI, as expected, since leptin related inversely to EHRI and BMI. Our study observed a similar causal relationship between BMI and ESA resistance. Some other studies have also shown an association between the level of MIS and ESA resistance level [42,43].

The association of lower BMI and the level of ESA resistance confirmed in our study can also be observed in the opposite direction since it has been previously shown that EPO can suppress obesity in both pre-clinical [44] and clinical studies [45].

This association can also be observed through the prism of lipid metabolism. Our study found L-FABP to be a significant predictor of ESA resistance. These results bringing BMI and L-FABP levels in association with ESA resistance may be related to the well-known fact that lipid metabolism plays a crucial role in erythropoiesis, particularly during the terminal stages of red blood cell development. Firstly, lipid metabolism is essential for providing the energy required during erythropoiesis. Fatty acids undergo oxidative phosphorylation to produce adenosine-triphosphate, which is crucial for the proliferation and maturation of erythroblasts [46]. In addition, the lipid composition of red blood cell membranes is critical for their survival and deformability. During erythropoiesis, specific lipids like phosphatidylcholine and phosphocholine are metabolized to maintain the proper membrane structure [47]. Also, the PHOSPHO1 gene, which encodes a phosphocholine phosphatase, is upregulated during terminal erythropoiesis. This gene is involved in the hydrolysis of phosphocholine to choline, which is necessary for energy balance and amino acid supply [47]. Finally, during the late stages of erythropoiesis, there is a shift from oxidative phosphorylation to glycolysis in erythroblasts. This shift is essential for the production of serine and glycine, which are important for red blood cell maturation [47] All of these processes highlight the intricate relationship between lipid metabolism and the development of red blood cells, ensuring they acquire the necessary properties to function effectively in the circulatory system.

This leads to the association between fatty acid metabolism and ESA resistance as a multifaceted relationship involving several metabolic and inflammatory pathways. Chronic inflammation, often seen in metabolic disorders, can interfere with EPO signaling. Proinflammatory cytokines can inhibit erythroid progenitor cells and disrupt iron metabolism, contributing to ESA resistance [48]. Abnormal lipid metabolism, particularly in conditions like obesity and metabolic syndrome, can affect EPO efficacy. For instance, EPO has been shown to regulate lipid metabolism by increasing fatty acid oxidation and promoting the browning of white adipose tissue [49]. There is also a link between lipid metabolism and insulin resistance, which can also impact EPO responsiveness. Insulin resistance can exacerbate inflammation and oxidative stress, further impairing EPO signaling [49]. In our study, we did not observe any relationship with the presence of diabetes mellitus, nor with the use of insulin. EPO also has protective effects on white adipose tissue, reducing inflammation and improving metabolic activity. However, in states of metabolic dysfunction, these protective effects may be diminished, leading to reduced EPO efficacy [49].

L-FABP has been shown previously as a biomarker for kidney damage and oxidative stress. Elevated levels of L-FABP in urine are associated with the progression of CKD. Since CKD patients often require EPO therapy to manage anemia, high L-FABP levels can indicate underlying kidney damage that may contribute to ESA resistance. Chronic inflammation and oxidative stress, a common occurrence in CKD, can also impair the effectiveness of EPO. Since L-FABP levels reflect the degree of oxidative stress, higher levels may correlate with increased inflammation, which is a known factor in ESA resistance. Studies have shown that EPO therapy can reduce urinary L-FABP levels in CKD patients, suggesting that effective EPO treatment may help mitigate some of the oxidative stress and kidney damage. However, in patients with high baseline L-FABP levels, the response to EPO might be less effective, indicating resistance [50].

L-FABP has also been previously shown to be hypoxia-induced and more prevalent in patients with anemia [51,52,53]. Also, the proximity of the hypoxia-sensitive site of production of renal L-FABP and EPO [17,18,19,20,54]. Although this pathophysiologic mechanism may be associated with local hypoxia and thus oxidative stress, in our study we did not find an association between the levels of investigated oxidative stress-related biomarkers and neither EHRI nor belonging to the ESA-resistant group.

L-FABP is involved in lipid metabolism, and disruptions in lipid metabolism can additionally affect erythropoiesis. Abnormal lipid profiles and metabolic dysregulation are common in CKD and can contribute to ESA resistance [50]. Understanding these connections can help in managing ESA resistance in CKD patients by addressing underlying inflammation, oxidative stress, and metabolic issues,

This study implemented a comprehensive approach to analyzing demographic, clinical, laboratory, and biomarker-related data, but has some limitations. Despite the fact that the statistical methods analyzed causal–consecutive relationships between the investigated parameters and ESA resistance, its observational design results in its limited ability to report specific causalities and to have a high benefit of randomization. In addition, the influence of hypertension as a comorbidity was not analyzed since all the patients receiving ESA therapy had hypertension, which was also a cause of CKD. This limitation of our analysis could increase the risk of overlooking the potential co-effect of the presence of hypertension on response to ESAs. Also, all the patients received a short-acting type of rHuEPO, which reduced the ability of this study to compare the proportions of ESA resistance between different types of rHuEPO. Both characteristics eliminated the potential effect of different modalities of ESA treatment and CKD of causes other than hypertension. However, these features do not necessarily represent a limitation if the results are observed only in the groups of patients with hypertension-caused CKD who receive short-acting rHuEPO. Due to the limitation set by the study protocol, we did not investigate more known oxidative stress-related biomarkers. Due to the same reason, we also did not analyze the level of insulin resistance, which may have the potential to be associated with lipid metabolism, a common connection with ESA resistance. Also, we did not analyze the type of vascular access in these patients and its effect on the level of ESA hyporesponsiveness. Additionally, while it includes 96 patients from a diverse population structure, this study was based on a convenience sampling method and analyzed data on patients from a single hemodialysis center, which might potentially limit the generalizability, geographical diversity of the patient population, and the applicability of the findings. Future research with a similar comprehensive analysis of several thrombo-inflammatory and oxidative stress-related biomarkers on the level of ESA hyporesponsiveness in ESRD patients treated with chronic hemodialysis should focus on larger samples and multicentricity in order to perceive the broader image of the common segments of pathogenesis of thrombo-inflammation, oxidative stress, and ESA hyporesponsiveness.

## 5. Conclusions

Patients expressing ESA resistance had lower BMI, serum iron level, iron saturation, and significantly less often used angiotensin-converting enzyme inhibitors and/or angiotensin-receptor blockers. These patients also expressed higher NLR, platelet-to-lymphocyte ratio, and MLR among the investigated cellular indices. Out of all the investigated thrombo-inflammatory and oxidative stress-related biomarkers, only L-FABP has been observed as a strong predictor of ESA resistance. High levels of this biomarker, as well as low BMI, were highly predictive for expressing ESA resistance.

## Figures and Tables

**Figure 1 diagnostics-14-02406-f001:**
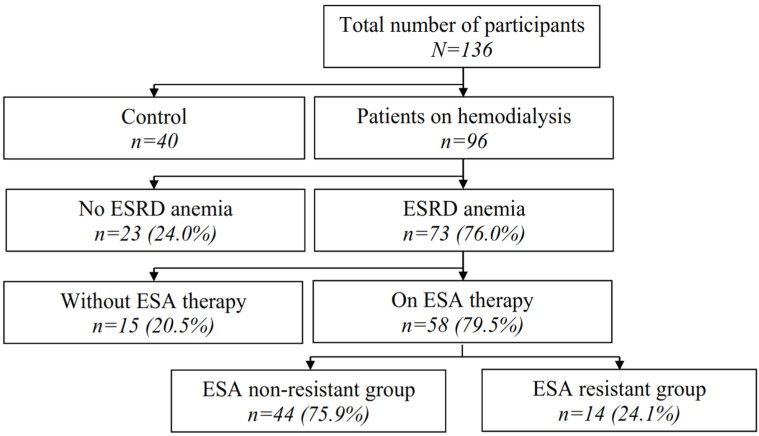
Patient selection flowchart (Legend: ESRD—end-stage renal disease; ESA—erythropoiesis-stimulating agents; EHRI—erythropoiesis-stimulating agents hyporesponsiveness index).

**Figure 2 diagnostics-14-02406-f002:**
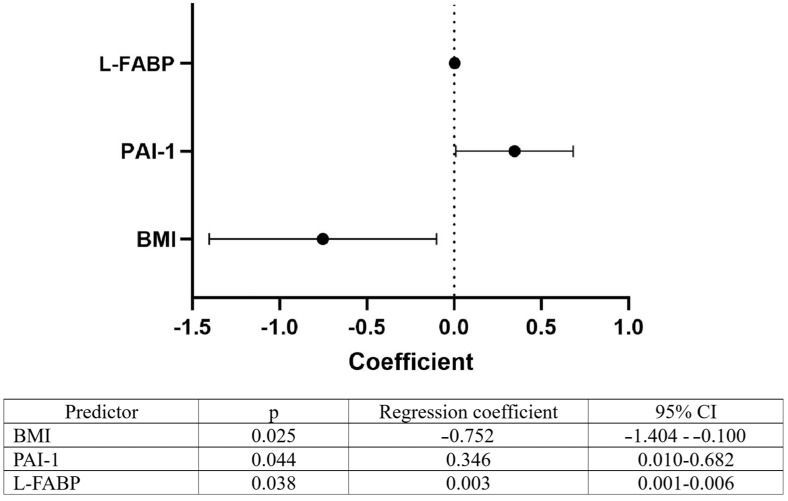
Predictors of EHRI in patients receiving rHuEPO (Legend: CI—confidence interval; BMI—body mass index; PAI-1—plasminogen activator inhibitor-1; L-FABP—L-type fatty acid binding protein).

**Figure 3 diagnostics-14-02406-f003:**
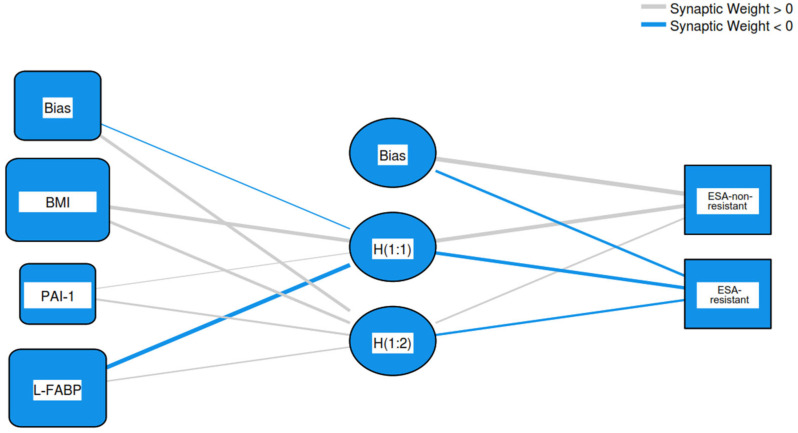
Artificial neural network architectural structure of the relationship between predictors and ESA resistance (Legend: BMI—body mass index; PAI-1—plasminogen activator inhibitor-1; L-FABP—L-type fatty acid binding protein; ESA—erythropoiesis-stimulating agents).

**Figure 4 diagnostics-14-02406-f004:**
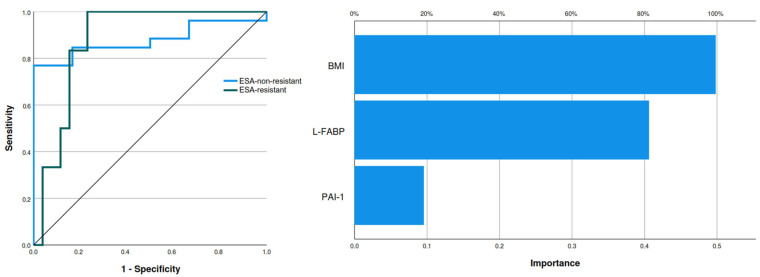
Area under the curve (**left**) and normalized importance analysis (**right**) of ESA resistance predicting model (Legend: ESA—erythropoiesis-stimulating agents; BMI—body mass index; L-FABP—L-type fatty acid binding protein; PAI-1—plasminogen activator inhibitor-1).

**Figure 5 diagnostics-14-02406-f005:**
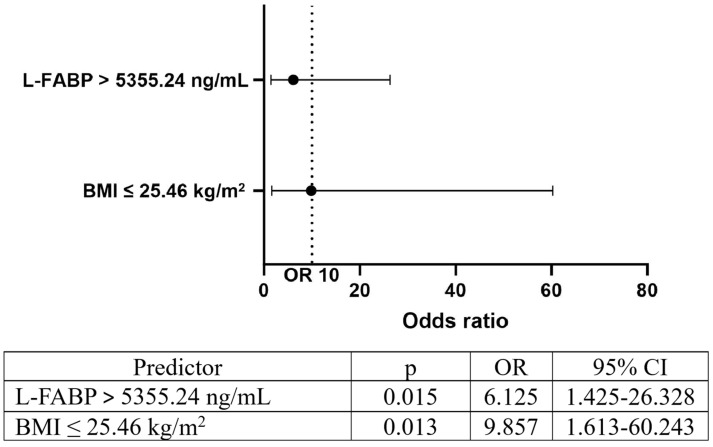
Predictors of EHRI in patients receiving rHuEPO (Legend: OR—odds ratio; CI—confidence interval; BMI—body mass index; PAI-1—plasminogen activator inhibitor-1; L-FABP—L-type fatty acid binding protein).

**Figure 6 diagnostics-14-02406-f006:**
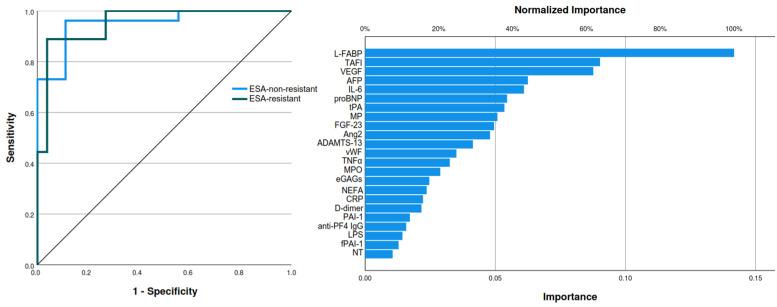
Area under the curve (**left**) and normalized importance analysis (**right**) of the investigated biomarkers on ESA resistance (Legend: L-FABP—L-type fatty acid binding protein; TAFI—thrombin activatable fibrinolysis inhibitor; VEGF—vascular endothelial growth factor; AFP—alpha-fetoprotein; IL-6—interleukin-6; proBNP—prohormone of brain natriuretic peptide; tPA—tissue-type plasminogen activator; MP—microparticles; FGF-23—fibroblast growth factor-23; Ang2—angiopoietin 2; ADAMTS-13—a disintegrin and metalloproteinase with a thrombospondin type 1 motif, member 13; vWF—von Willebrand factor; TNF-α—tumor necrosis factor-alpha; MPO—myeloperoxidase; eGAGs—endogenous glycosaminoglycans; NEFA—non-esterified fatty acids; CRP—C-reactive protein; PAI-1—plasminogen activator inhibitor-1; PF4—platelet factor 4; IgG—immunoglobulin G; LPS—lipopolysaccharide; fPAI-1—functional plasminogen activator inhibitor-1; NT- nitrotyrosine).

**Table 1 diagnostics-14-02406-t001:** Levels of analyzed biomarkers in ESRD and control group.

Biomarkers	Mean ± SD/Med (IQR)		*p*
	ESRD	Control	
D-Dimer (ng/mL)	1249.08 (550.37–2361.31)	67.10 (58.36–82.03)	<0.001
PAI-1 (ng/mL)	15.40 (9.39–28.48)	3.26 (1.56–7.98)	<0.001
TAFI (%)	103.35 ± 10.52	79.38 ± 2.73	<0.001
vWF (%)	85.11 (77.38–92.50)	67.10 (58.36–82.03)	0.001
CRP (ng/mL)	6206.73 (3440.02–9530.94)	3049.78 (2383.59–3343.22)	0.005
anti-PF4 IgG (OD at 450 nm)	0.05 (0.03–0.12)	0.04 (0.02–0.05)	0.017
eGAGs (μg/mL)	0.89 (0.84–0.93)	0.26 (0.08–0.47)	<0.001
tPA (ng/mL)	4.43 (1.19–6.95)	0.37 (0.26–0.46)	<0.001
Microparticles (nM)	21.41 (11.87–34.68)	11.59 (10.47–13.65)	0.025
Functional PAI-1 (IU/mL)	7.02 (3.17–15.10)	3.21 (2.39–5.14)	0.009
AFP (ng/mL)	2.51 ± 1.77	2.55 ± 0.56	0.886
MPO (ng/mL)	100.70 (58.30–160.73)	27.31 (24.67–39.89)	<0.001
Nitrotyrosine (nM)	131.37 (25.94–1067.89)	113.88 (45.89–157.45)	0.482
ADAMTS-13 (IU/mL)	0.83 ± 0.19	1.25 ± 0.14	<0.001
Angiopoietin 2 (mg/mL)	1476.52 (1107.19–1781.69)	304.95 (285.15–374.96)	<0.001
NT-proBNP (ng/mL)	28.55 (13.32–36.35)	0.00 (0.00–0.00)	<0.001
L-FABP (ng/mL)	4907.63 ± 1786.20	586.48 ± 655.97	<0.001
FGF-23 (ng/mL)	0.13 (0.00–10.13)	4.49 (0.00–23.63)	0.324
IL-6 (pg/mL)	2.33 (0.00–4.97)	1.60 (0.00–2.89)	0.217
LPS (ng/mL)	2.11 (1.95–2.28)	2.80 (2.52–3.20)	<0.001
NEFA (mmol/L)	0.56 (0.43–0.86)	0.90 (0.58–1.16)	0.017
TNF-α (pg/mL)	9.32 (6.21–12.00)	0.59 (0.00–2.19)	<0.001
VEGF (pg/mL)	37.94 (28.66–46.56)	10.88 (3.33–19.82)	<0.001

Legend: SD—standard deviation; IQR—interquartile range; ESRD—end-stage renal disease; PAI-1—plasminogen activator inhibitor-1; TAFI—thrombin activatable fibrinolysis inhibitor; vWF—von Willebrand factor; CRP—C-reactive protein; PF4—platelet factor 4; IgG—immunoglobulin G; OD—optical density; eGAGs—endogenous glycosaminoglycans; tPA—tissue-type plasminogen activator; AFP—alpha-fetoprotein; MPO—myeloperoxidase; ADAMTS-13—a disintegrin and metalloproteinase with a thrombospondin type 1 motif, member 13; NT-proBNP—N-terminal prohormone of brain natriuretic peptide; L-FABP—L-type fatty acid binding protein; FGF-23—fibroblast growth factor-23; IL-6—interleukin-6; LPS—lipopolysaccharide; NEFA—non-esterified fatty acids; TNF-α—tumor necrosis factor-alpha; VEGF—vascular endothelial growth factor.

**Table 2 diagnostics-14-02406-t002:** Comparison of clinical, demographic, and laboratory characteristics between the patients in the ESA-non-resistant group and ESA-resistant group.

Variables	Mean ± SD/Med (IQR)		*p*
	ESA-Non-Resistant Group (%)	ESA-Resistant Group (%)	
Age (years)	67.00 (51.00–74.00)	66.00 (41.50–73.50)	0.919
MAP (mmHg)	89.91 ± 12.29	95.31 ± 14.58	0.178
BMI (kg/m^2^)	31.13 ± 8.13	24.63 ± 4.58	0.038
BSA (m^2^)	1.99 ± 0.35	1.82 ± 0.25	0.207
CII	6.00 (4.00–8.00)	6.50 (6.00–8.00)	0.288
HD duration (months)	50.00 (21.50–111.00)	36.00 (10.75–87.75)	0.468
BW gain (kg)	4.06 ± 2.60	4.19 ± 2.86	0.870
Kt/V	1.52 (1.39–1.81)	1.58 (1.34–1.84)	0.769
URR (%)	74.68 (70.31–77.27)	75.00 (68.43–77.66)	0.956
Serum iron (μg/dL)	60.00 (50.00–75.25)	46.50 (35.75–61.50)	0.023
Iron saturation (%)	28.50 (23.75–34.75)	23.00 (16.25–28.25)	0.032
TF (mg/dL)	160.55 ± 26.90	160.14 ± 27.53	0.962
FER (ng/mL)	713.36 ± 326.54	766.14 ± 436.03	0.633
ALP (IU/L)	84.00 (62.00–109.00)	85.00 (71.75–122.25)	0.504
Serum albumin (g/dL)	3.85 ± 0.36	3.73 ± 0.26	0.266
Serum Ca (mg/dL)	8.50 (8.00–9.10)	8.70 (8.30–9.00)	0.446
Serum P (mg/dL)	5.70 (4.50–7.00)	5.25 (4.58–5.80)	0.409
CaxP (mg^2^/dL^2^)	46.56 (40.80–62.38)	46.42 (39.26–53.60)	0.704
Ca corrected for albumin (md/dL)	8.70 ± 0.61	8.89 ± 0.67	0.325
iPTH (pg/mL)	511.50 (295.25–681.25)	392.50 (240.25–542.50)	0.150
Total serum protein (g/dL)	7.02 ± 0.46	6.86 ± 0.44	0.280
Potassium (mmol/L)	4.62 ± 0.52	4.65 ± 0.45	0.864
Vitamin D (ng/mL)	33.90 ± 19.13	25.61 ± 10.97	0.202
Neutrophil-to-lymphocyte ratio	3.00 (2.00–3.73)	4.12 (3.75–6.13)	0.005
Derived neutrophil-to-lymphocyte ratio	1.72 (1.25–2.28)	2.14 (1.93–2.36)	0.105
Platelet-to-lymphocyte ratio	142.78 (115.79–181.00)	191.67 (149.23–291.36)	0.038
Monocyte-to-lymphocyte ratio	0.44 (0.33–0.57)	0.62 (0.47–0.90)	0.024
Platelet-to-monocyte ratio	333.75 (260.00–410.00)	326.67 (225.25–450.86)	0.846
Hemoglobin-to-platelet ratio	0.05 (0.04–0.06)	0.04 (0.03–0.05)	0.153
Hemoglobin-to-lymphocyte ratio	8.16 ± 3.92	9.28 ± 3.39	0.345
Neutrophil-to-eosinophil ratio	14.75 (9.56–28.38)	23.50 (9.21–32.00)	0.714
Lymphocyte-to-RBC ratio	0.47 ± 0.20	0.39 ± 0.17	0.175
Neutrophil-to-RBC ratio	1.29 (0.95–1.71)	1.43 (1.26–1.81)	0.266
Monocyte-to-RBC ratio	0.21 ± 0.09	0.22 ± 0.10	0.565
Neutrophil-to-platelet ratio	0.02 ± 0.01	0.02 ± 0.01	0.697
SII	595.54 (421.24–928.06)	832.40 (554.85–1599.14)	0.128
SIRI	1.77 (1.05–2.70)	2.68 (1.78–4.23)	0.059

Legend: SD—standard deviation; IQR—interquartile range; ESA—erythropoiesis-stimulating agents; MAP—mean arterial pressure; BMI—body mass index; BSA—body surface area; CII—Charlson comorbidity index; HD—hemodialysis; BW—body weight; Kt/V—fractional urea clearance; URR—urea reduction ratio; TF—transferrin; FER—ferritin; ALP—alkaline phosphatase; Ca—calcium; P—phosphorus; iPTH—intact parathyroid hormone; RBC—red blood cell ratio; SII—systemic immune-inflammation index; SIRI—systemic inflammation response index.

**Table 3 diagnostics-14-02406-t003:** Comparison of comorbidity distribution between the patients in the ESA-non-resistant group and ESA-resistant group.

Comorbidities		ESA-Non-Resistant Group (%)	ESA-Resistant Group (%)	*p*
Coronary vascular disease	Yes	71.4	28.6	0.591
	No	65.1	57.1	
Chronic pulmonary disease	Yes	68.0	32.0	0.249
	No	81.3	18.8	
Ischemic heart disease	Yes	66.7	33.3	0.240
	No	80.6	19.4	
Diabetes mellitus	Yes	82.1	17.9	0.107
	No	61.1	38.9	
Chronic heart failure	Yes	79.2	20.8	0.757
	No	72.7	27.3	
Malignancy	Yes	80.0	20.0	>0.999
	No	74.5	25.5	
Hypercholesterolemia	Yes	66.7	33.3	0.629
	No	76.5	23.5	
Hyperlipidemia	Yes	93.7	6.3	0.083
	No	68.3	31.7	
Iron deficiency anemia	Yes	86.7	13.3	0.312
	No	71.4	28.6	
Coagulopathy	Yes	100.0	0.0	>0.999
	No	75.0	25.0	

Legend: ESA—erythropoiesis-stimulating agents.

**Table 4 diagnostics-14-02406-t004:** Comparison of applied pharmacotherapy at the beginning of the follow-up period between the patients in the ESA-non-resistant group and ESA-resistant group.

Therapy Taken		ESA-Non-Resistant Group (%)	ESA-Resistant Group (%)	*p*
Anti-inflammatory drugs	Yes	58.3	41.7	0.144
	No	80.0	20.0	
Anti-phosphate agents	Yes	71.4	28.6	0.375
	No	81.8	18.2	
Beta-blockers	Yes	65.6	34.4	0.051
	No	88.0	12.0	
ACEIs/ARBs	Yes	100.0	0.0	0.013
	No	67.4	32.6	
Statins	Yes	80.0	20.0	0.313
	No	68.2	31.8	
Antithrombotic drugs	Yes	64.3	35.7	0.297
	No	79.1	20.9	
Iron supplementation	Yes	60.0	40.0	0.059
	No	83.8	16.2	
Folic acid supplementation	Yes	79.3	20.7	0.490
	No	71.4	28.6	
Vitamin B12 supplementation	Yes	60.0	40.0	0.161
	No	81.0	19.0	
Vitamin C supplementation	Yes	78.6	21.4	0.589
	No	72.4	27.6	
Vitamin D supplementation	Yes	76.9	23.1	0.812
	No	74.2	25.8	

Legend: ESA—erythropoiesis-stimulating agents; ACEIs—angiotensin-converting enzyme inhibitors; ARBs—angiotensin II receptor blockers.

**Table 5 diagnostics-14-02406-t005:** Comparison of the levels of investigated biomarkers between the patients in the ESA-non-resistant group and ESA-resistant group.

Biomarkers	Mean ± SD/Med (IQR)		*p*
	ESA-Non-Resistant Group (%)	ESA-Resistant Group (%)	
D-Dimer (ng/mL)	1470.59 (670.77–3245.40)	816.73 (267.10–2381.43)	0.217
PAI-1 (ng/mL)	15.69 (9.40–25.51)	16.68 (6.77–33.19)	0.990
TAFI (%)	103.68 ± 10.53	103.76 ± 6.16	0.974
vWF (%)	85.98 (79.83–93.99)	87.65 (82.84–95.02)	0.577
CRP (ng/mL)	7012.73 ± 4085.30	4903.08 ± 3255.45	0.125
anti-PF4 IgG (OD at 450 nm)	0.04 (0.03–0.10)	0.05 (0.03–0.21)	0.795
eGAGs (μg/mL)	0.89 (0.85–0.94)	0.86 (0.83–0.90)	0.314
tPA (ng/mL)	3.57 (1.19–6.33)	7.63 (0.90–10.12)	0.511
Microparticles (nM)	21.88 (13.22–30.71)	12.69 (6.55–39.21)	0.158
Functional PAI-1 (IU/mL)	7.04 (4.31–14.51)	9.98 (0.01–20.85)	0.434
AFP (ng/mL)	2.38 ± 1.86	2.62 ± 1.13	0.592
MPO (ng/mL)	103.23 ± 59.21	121.55 ± 61.53	0.363
Nitrotyrosine (nM)	120.21 (21.56–1354.12)	163.03 (43.74–789.80)	0.721
ADAMTS-13 (IU/mL)	0.85 ± 0.15	0.75 ± 0.17	0.049
Ang2 (mg/mL)	1494.95 (1091.67–1756.45)	1302.48 (1179.95–1663.00)	0.665
proBNP (ng/mL)	30.53 ± 18.50	25.18 ± 15.92	0.414
FABP (ng/mL)	4498.20 ± 1668.68	5603.93 ± 1450.46	0.054
FGF-23 (ng/mL)	1.10 (0.01–8.34)	0.02 (0.01–8.59)	0.569
IL-6 (pg/mL)	1.56 (0.01–4.89)	4.12 (2.40–6.87)	0.078
LPS (ng/mL)	2.09 (1.89–2.26)	2.16 (2.03–2.24)	0.542
NEFA (mmol/L)	0.73 ± 0.38	0.62 ± 0.23	0.397
TNFa (pg/mL)	8.44 ± 4.74	11.32 ± 4.17	0.080
VEGF (pg/mL)	35.53 ± 11.08	41.37 ± 10.02	0.126

Legend: SD—standard deviation; IQR—interquartile range; EHRI—erythropoiesis-stimulating agents hyporesponsiveness index; PAI-1—plasminogen activator inhibitor-1; TAFI—thrombin activatable fibrinolysis inhibitor; vWF—von Willebrand factor; CRP—C-reactive protein; PF4—platelet factor 4; IgG—immunoglobulin G; OD—optical density; eGAGs—endogenous glycosaminoglycans; tPA—tissue-type plasminogen activator; AFP—alpha-fetoprotein; MPO—myeloperoxidase; ADAMTS-13—a disintegrin and metalloproteinase with a thrombospondin type 1 motif, member 13; NT-proBNP—N-terminal prohormone of brain natriuretic peptide; L-FABP—L-type fatty acid binding protein; FGF-23—fibroblast growth factor-23; IL-6—interleukin-6; LPS—lipopolysaccharide; NEFA—non-esterified fatty acids; TNF-α—tumor necrosis factor-alpha; VEGF—vascular endothelial growth factor.

**Table 6 diagnostics-14-02406-t006:** Distribution of the number of ESA-non-resistant and ESA-resistant patients between BMI and L-FABP groups.

Predictors		ESA-Non-Resistant Group (%)	ESA-Resistant Group (%)	*p*
BMI	≤25.46 kg/m^2^	53.8	46.2	0.011
	>25.46 kg/m^2^	92.0	8.0	
L-FABP	≤5355.24 ng/mL	53.3	46.7	0.023
	>5355.24 ng/mL	87.5	12.5	

Legend: ESA—erythropoiesis-stimulating agents; BMI—body mass index; L-FABP—L-type fatty acid binding protein.

## Data Availability

Dataset available on request from the authors. The raw data supporting the conclusions of this article will be made available by the authors on request.

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
