# Peer review of "Severe Hyporesponsiveness to Erythropoiesis-Stimulating Agents in Patients on Chronic Hemodialysis—Reconsidering the Relationship with Thrombo-Inflammation and Oxidative Stress"

_diagnostics, 2024, doi:10.3390/diagnostics14212406_

Round 1

Reviewer 1 Report

Comments and Suggestions for Authors

- Sample Size and Diversity: While the study includes 96 patients, the sample size might still be considered small for generalizability. Additionally, it appears to come from a single center, potentially limiting the diversity of the patient population and the applicability of the findings.

- Definition of ESA Hyperresponsiveness's method for defining ESA hyporesponsiveness may lack clarity. Differentiating between mild and severe hyporesponsiveness, and providing more explicit cut-off values or criteria, could enhance understanding.

- Weakness in Addressing Limitations: Although limitations are mentioned, they might not be explored in depth. Acknowledging the potential impact of these limitations on the study outcomes is important for transparency

Author Response

Dear Reviewer,

We appreciate the time and effort that you dedicated to providing feedback on our manuscript “Severe Hyporesponsiveness to Erythropoiesis-Stimulating Agents in Patients on Chronic Hemodialysis - Reconsidering the Relationship with Thrombo-inflammation and Oxidative Stress” and are grateful for the insightful comments on and valuable improvements to our paper.

We have incorporated most of your suggestions. Therefore, we believe that the revised version of our manuscript addresses your requests correctly and that, as such, it will be suitable for publication.

Please see below a point-by-point response to your comments.

Comment 1: Sample Size and Diversity: While the study includes 96 patients, the sample size might still be considered small for generalizability. Additionally, it appears to come from a single center, potentially limiting the diversity of the patient population and the applicability of the findings.

Response 1: Thank you for your valuable comment. We entirely agree with the suggestion and have entered addition to the limitations section of the Discussion (Lines 521-524) emphasizing the potential limiting effect of sample size and diversity on the results obtained in this study.

Comment 2: Definition of ESA Hyperresponsiveness's method for defining ESA hyporesponsiveness may lack clarity. Differentiating between mild and severe hyporesponsiveness, and providing more explicit cut-off values or criteria, could enhance understanding.

Response 2: Thank you for your comment. In the existing literature, no exact rule has been proposed and accepted in general in order to define ESA-hyporesponsiveness and ESA-resistance. Different studies in the past used their own methods and criteria to define these terms (e.g. doi: 10.5812/numonthly.45003; doi: 10.1053/j.ajkd.2008.12.040; doi: 10.1159/000500921). In our study, we defined "ESA-resistance as "ESA-severe hyporesponsiveness" (as it has already been emphasized at the end of the sub-section "2.2 Samples and data collection"). We explained the process of defining these terms in a more detail in order to improve the level of understanding (Lines 123-127).

Comment 3: Weakness in Addressing Limitations: Although limitations are mentioned, they might not be explored in depth. Acknowledging the potential impact of these limitations on the study outcomes is important for transparency.

Response 3: We are grateful for and agree with this constructive suggestion. Additional explanations of potential impact of our study's limitations have been added to the limitations section of the Discussion (Lines 502-529).

On behalf of all the contributors, thank you once again for your time and effort in reviewing of our manuscript.

Reviewer 2 Report

Comments and Suggestions for Authors

I really liked this study, the only thing I didn't understand was why in table 1 some measures are described using median and IQR and others M+-SD, for example AFP (ng/mL) was definitely not normally distributed and in this case Md [IQR] should be used.

And I have a question for the authors of the study - did this analysis use training and testing data sets, because not using such a division can overestimate the prediction models.

Author Response

Dear Reviewer,

We appreciate the time and effort that you dedicated to providing feedback on our manuscript “Severe Hyporesponsiveness to Erythropoiesis-Stimulating Agents in Patients on Chronic Hemodialysis - Reconsidering the Relationship with Thrombo-inflammation and Oxidative Stress” and are grateful for the insightful comments on and valuable improvements to our paper.

We have incorporated most of your suggestions. Therefore, we believe that the revised version of our manuscript addresses your requests correctly and that, as such, it will be suitable for publication.

Please see below a point-by-point response to your comments.

Comment 1: I really liked this study, the only thing I didn't understand was why in table 1 some measures are described using median and IQR and others M+-SD, for example AFP (ng/mL) was definitely not normally distributed and in this case Md [IQR] should be used.

Response 1: Thank you for your comment. In our manuscript, there is no statement indicating that the distribution of values representing levels of AFP is significantly different from normal distribution. Even after repeating the normality tests on all variables made by a biostatistics expert, we did not observe normal distribution of AFP-related data. Therefore, it is our opinion that mean value and standard deviation should be used in order to represent the central tendency value and distribution of AFP levels in this cohort.

Comment 2: And I have a question for the authors of the study - did this analysis use training and testing data sets, because not using such a division can overestimate the prediction models.

Response 2: Thank you for your valuable comment. In both of the multilayer perceptron artificial neural network modeling analyses we implemented the division to training and testing data sets. The appropriate description of the proportion of participants allocated to both of the divisions have been added to the Results section of the manuscript (Lines 274, 275, 326, and 327).

On behalf of all the contributors, thank you once again for your time and effort in reviewing of our manuscript.